# Sarcopenia and Liver Cirrhosis—Comparison of the European Working Group on Sarcopenia Criteria 2010 and 2019

**DOI:** 10.3390/nu12020547

**Published:** 2020-02-20

**Authors:** Julia Traub, Ina Bergheim, Martin Eibisberger, Vanessa Stadlbauer

**Affiliations:** 1Department of Clinical Medical Nutrition, University Hospital Graz, 8036 Graz, Austria; 2Department of Nutritional Sciences, RF Molecular Nutritional Science, University Vienna, 1090 Vienna, Austria; 3Department of Neuroradiology, Vascular and Interventional Radiology, Medical University of Graz, 8036 Graz, Austria; 4Department of Internal Medicine Division of Gastroenterology und Hepatology Medical University of Graz, Auenbruggerplatz 15, 8036 Graz, Austria

**Keywords:** liver cirrhosis, sarcopenia, consensus, muscle strength

## Abstract

The European Working group on Sarcopenia in Older People recently updated the diagnostic criteria for sarcopenia. It is yet unclear how these modified criteria influence the rate of diagnosis in high risk populations, such as liver cirrhosis. We therefore assessed if the new diagnostic criteria for sarcopenia impacts on sarcopenia prevalence in liver cirrhosis. Within two years 114 cirrhotic patients were prospectively enrolled in the study. Sarcopenia was determined by muscle strength (handgrip strength), muscle mass (lumbal muscle index) and muscle performance (gait speed). Using the 2019 definition, the rate of pre-sarcopenia was significantly lower (30.7% versus 3.5%) due to the different starting points (2010 muscle mass, 2019 muscle strength) and cut-off values (muscle strength). The change in diagnostic criteria for sarcopenia drastically influences the rate of pre-sarcopenia diagnosis in cirrhotics. To evaluate, which diagnostic criteria should be chosen to diagnose sarcopenia in liver cirrhosis patients, prospective studies are needed.

## 1. Introduction

The liver is the central organ in nutrient metabolism and has many important metabolic functions [1]. In patients with chronic liver diseases, these metabolic functions are impaired resulting in a variety of nutritional disorders, such as protein-energy-malnutrition (PEM) or muscle abnormalities [2]. Among a wide variety of muscle abnormalities in patients with liver cirrhosis [3], sarcopenia is described as the most common form [4]. Indeed, sarcopenia is prevalent in 30–70% of cirrhotic patients [5] with a higher prevalence in men (61.6%) than in woman (36%) [6]. So far, there is no gold standard for the diagnoses of sarcopenia in liver cirrhosis patients. General criteria for the diagnosis of sarcopenia are not universally accepted and several different definitions coexist. The European Working Group on Sarcopenia in Older People (EWGSOP) provided consensus criteria for the diagnosis of sarcopenia using muscle mass, muscle strength and muscle performance as a practical clinical definition [7]. In 2019, a revised definition was published [8]. However, up to now it remains to be determined how these criteria impact diagnosis of sarcopenia different patient populations. Therefore, we evaluated if the prevalence of sarcopenia in patients with liver cirrhosis differs when using the 2010 in comparison to the 2019 sarcopenia consensus definition of EWGSOP (Table 1).

## 2. Materials and Methods

The study was approved by the research ethics committee of the Medical University of Graz (29-280 ex 16/17) and is registered at clinicaltrials.gov (NCT03080129). The study was conducted after informed consent according to the principles of the Declaration of Helsinki. For this prospective cohort study 209 liver cirrhosis patients were screened between April 2017 and January 2019 at the Department of Gastroenterology and Hepatology of the Medical University of Graz. Hospitalized men or woman over the age of 18 years with clinical / radiological / histological diagnosis of cirrhosis and a CT/MRI (computed tomographic / magnetic resonance imaging) scan within +/−1 month of the baseline study visit were included in the study. Exclusion criteria were defined as hepatic encephalopathy > grade 2 and/or other cognitive disorder not allowing for informed consent, hepatocellular carcinoma stage BCLC C or D, current intake of antibiotics and any other condition or circumstance, which, in the opinion of the investigator, would affect the patient´s ability to participate in the protocol. We assessed the absence of sarcopenia (no sarcopenia group) or the presence of pre-sarcopenia or sarcopenia/severe sarcopenia using both 2010 and 2019 definitions [7,8].

To diagnose sarcopenia, muscle mass, strength and functional performance were evaluated: Muscle mass was assessed by the analysis of routinely performed CT or MRI scans of the abdomen using the image analyzing software SliceOmatic 5.0 from Tomovison. The lumbar skeletal muscle index was determined by segmentation of the cross-sectional muscle area at L3 level; including the M.psoas, M.paraspinalis, M.transversus abdominis, M. obliquus internus and externus, M. rectus abdominus and M.quadratus lumborum) divided by the height squared being than normalize to the stature [9,10,11,12]. For CT-pictures, automatically segmentation using a Hounsfield unit (HU) threshold of −30 to +150 was performed. For MRI-pictures, manual segmentation on axial T2-TSE sequences was performed. The cut-offs for low muscle mass are not well defined in both consensus definitions. In our study we therefore decided to use validated liver disease specific cut-offs from a meta-analysis published in 2017 [6]. Low muscle mass was defined as L3-muscle-area of ≤52.4 cm^2^/m^2^ in male and ≤38.5 cm^2^/m^2^ in female cirrhotic patients. [6] Muscle strength was assessed by measuring hand grip strength using a hydraulic dynamometer (Jamar Hydraulic Handdynamometer) applying cut-off values of the EWGSOP 2010 and 2019. Muscle function or functional performance were evaluated by 4-m gait speed with <0.8 m/s as cut-off for decreased physical performance, defined by the EWGSOP 2010 and 2019 definition criteria [7,8].

## 3. Results

### 3.1. Baseline Characteristics

209 patients were screened for the study. 95 (45.5%) were excluded: 17 (17.9%) had no CT/MRI within the last 2 months, in 7 (7.4%) diagnosis of cirrhosis could not be confirmed, 24 (25.3%) fulfilled at least one exclusion criteria, 17 (17.9%) had missing data and 30 (31.6%) did not consent to be enrolled in the study. Accordingly, 114 patients were included in the study. In 90 cases (78.9%) a CT-scan and in 24 cases (21.1%) a MRI scan was used for diagnosis of muscle mass. The median age of the study population was 65 years (21–87). As expected, only 28 out of 114 included patients were women (24.6%). Alcohol was the main cause of liver disease in 57.9% of the patient population followed by non-alcoholic steatohepatitis (25.4%) and viral hepatitis (16.7%). 44.7% had a Child A cirrhosis and the median Model for End-stage Liver Disease (MELD) score was 10.8. Detailed baseline characteristics of the study population are shown in Table 2.

### 3.2. Frequency of Sarcopenia Applying the EWGSOP 2010 and 2019 Criteria

Based on the 2010 definition, 38/114 (33.3%) patients had no sarcopenia, 35/114 (30.7%) suffered from pre-sarcopenia and 41/114 (36%) from sarcopenia. Of the 41 patients with sarcopenia, 17 were classified as severe sarcopenic (14.1% of the whole cohort). With the 2019 definition, significantly more patients (91/114, 79.8%) were diagnosed as non-sarcopenic, whereas only 4/114 (3.5%) were diagnosed with pre-sarcopenia and 19/114 (16.7%) with sarcopenic (*p* < 0.0001). Twelve of the 19 patients with sarcopenia were classified as being severe sarcopenic (10.5% of the whole cohort). Figure 1 quantifies and visualizes the flow between the groups in a Sankey diagram.

When applying the 2010 definition, significantly more men were diagnosed as pre-sarcopenic (80% of 35, *p* = 0.042) and sarcopenic (87.8% of 41, *p* = 0.003) compared to the non-sarcopenic group. Further, BMI was significantly higher in the non-sarcopenic group compared to the pre-sarcopenic (*p* = 0.000) and sarcopenic group (*p* = 0.001). Using the 2019 definition, also significantly more men were diagnosed as sarcopenic (94.7% of 19, *p* = 0.038) compared to the no sarcopenic group, whereas the BMI showed now differences between the groups. To visualize and quantify the changes in sarcopenia diagnosis in women and men using the EWGSOP 2010 and 2019 criteria, separate Sankey diagrams for gender were generated (Figure 2A,B).

### 3.3. Individual Components of the EWGSOP Diagnostic Criteria

According to the EWGSOP 2010 definition, for hand grip strength cut-offs may or may not be adjusted to BMI and gender (Table 3) whereas in the 2019 definition, only dichotomous cut-offs with <27 kg for men and <16 kg for woman are suggested. Using the dichotomous cut-off values for hand grip strength (HGS) from the sarcopenia consensus 2010 without BMI-matching (<30 kg in men and <20 kg in woman), 44/114 (38.9%) patients were diagnosed to have a reduced muscle strength. When applying the optional BMI-matching, 48/114 (42.1%) were diagnosed with reduced hand grip strength. With the more stringent cut-off values from 2019, only 23/114 (20.2%) patients from our study population were diagnosed with reduced muscle strength, resulting in a reduction of patients diagnosed with reduced muscle strength by 52% (*p* < 0.0001). When using the 2019 criteria, 35 patients (30.7%) with reduced muscle mass but preserved muscle strength were not diagnosed as pre-sarcopenia any more compared to the 2010 criteria due to the different starting point of the definition. Four patients (3.5%) who were not diagnosed with sarcopenia according to the 2010 criteria shifted to the pre-sarcopenia group with the 2019 criteria because of a reduction in muscle strength but not in muscle mass.

For muscle mass, the cut-offs we used [6] were recently challenged by a study, which defines low muscle mass as <39 cm^2^/m^2^ in woman and <50 cm^2^/m^2^ in men [13]. Using these new cut-offs, 2/38 (both female) patients who were classified as having a normal muscle mass before, were reclassified as low muscle mass and 12/76 (only male) patients with previous low muscle mass were reclassified to the group with normal muscle mass since the cut-off increased by 1.3% for women (38.5/39 cm^2^/m^2^) and decreased by 4.5% for men (52.4/50 cm^2^/m^2^). Using the updated cut-off values for both the 2010 and 2019 resulted in an equal reduction of sarcopenia diagnoses (41 vs. 37 with the 2010 definition (9.8%) and 19 vs. 16 people with the 2019 definition (15.8%); NS). We also analyzed the impact of different imaging modalities: we did not observe any differences in diagnosis frequency of sarcopenia between CT and MR images. For gait speed, the cut-off values (<0.8 m/s) did not changed between the definitions.

## 4. Discussion

As sarcopenia is difficult to diagnose, several definitions are presently used. The recommendations of the EWGSOP for the diagnoses of sarcopenia are among the most frequently used offering an algorithm and cut-off values to determine sarcopenia. In the present study, we were able to show in patients with liver cirrhosis that with the updated 2019 consensus definition, pre-sarcopenia is significantly less often diagnosed compared to the 2010 definition.

In 2010, the EWGSOP designed a staging concept which graduates the presence of sarcopenia in pre-sarcopenia, sarcopenia and severe sarcopenia. Pre-sarcopenia is defined as the presence of low muscle mass without impact on muscle strength or physical performance. Sarcopenia is defined as low muscle mass and additionally low muscle strength or impaired muscle function. In severe sarcopenia, muscle mass, muscle strength and muscle function are impaired [4]. To help in the practical application, cut-off values for muscle mass, strength and performance were provided [7]. In 2019, a revised consensus with modified criteria was published [8]. Herein, development of sarcopenia was recognized to begin earlier in life [14] and muscle strength was replacing the role of muscle mass as the entry determinant in the 2019 definition with more stringent cut-off values (2010 vs. 2019: <30 kg vs. <27 kg for men and <20 kg vs. <16 kg for woman) [15]. This change was made because muscle strength and function showed better association with outcome compared to muscle mass [15]. In the 2019 definition, pre-sarcopenia was therefore defined as the presence of low muscle strength without impact on muscle mass/quality or muscle function. Sarcopenia was defined as low muscle strength and additionally low muscle mass or quality. For severe sarcopenia muscle mass, muscle strength and muscle function needed to be impaired. With this change in the diagnostic algorithm, case-finding in research and in the clinical setting should be improved [8]. Our study shows that this change in the starting point of the definition of sarcopenia (muscle strength versus muscle mass) and in the cut-off values leads to significantly different results in the rate of diagnosis of sarcopenia in liver cirrhosis.

Additionally, cut-off values for muscle mass, muscle strength and muscle function were varying in literature. Regarding cut-offs for muscle mass, in the 2010 consensus definition the general statement was made, that a cut-off value of below 2 standard deviations of the mean reference value of healthy young adults should be used, without referencing any specific study or cut-off points for CT/MRI. In the 2019 definition again cut-off values were suggested to be derived from the general healthy population, referencing a study in 420 healthy Caucasians [16], but the necessity of disease specific cut-offs was discussed as well. Disease specific cut-offs for muscle mass were established for liver cirrhosis patients from a meta-analysis [6] but recently challenged by the results of a multicenter study in patients with end-stage liver disease, which suggest slightly different cut-offs [13]. Currently there is no agreement between scientific societies which of these cut-offs should be used [17,18]. Our data show that even slight changes in these cut-offs lead to relevant changes in the frequency of the diagnosis of low muscle mass, especially when changes go in different directions for men and women.

To measure muscle mass in clinical practice both, CT and MRI pictures are used. While the- prognostic value of CT images is well validated [19], there are concerns, that the difference in imaging modality may affect the results of the assessment of skeletal muscle for example by the lack of standardized assessment protocol for MRI pictures [20]. Some studies however suggest that both imaging modalities can be used equivalently [21,22]. When comparing results from CT and MRI pictures in our study cohort, there was no difference in the detection rate of reduced muscle mass. However, we did not have the opportunity to compare CT and MRI images in the same patients, since we used imaging studies that were routinely performed.

In our study we observed that the 2010 definition identifies more sarcopenia cases in male patients compared to the 2019 definition. In patients with cirrhosis, muscle strength seems to be preserved longer, while muscle mass is already reduced, leading to significant difference in sarcopenia diagnosis rates when using the 2019 definition, in which muscle strength is the starting point. A couple of studies already compared the 2010 and the 2019 criteria in elderly and in in different disease cohorts. In contrast to our data in liver cirrhosis, in patients with peritoneal dialysis the definition from 2010 identified less patients as sarcopenic compared to the 2019 definition (4% versus 10%) [23]. In this study, however, muscle mass was assessed using dual energy x-ray absorptiometry (DEXA), which might also be a reason for the discrepant results compared to liver cirrhosis [23]. Large differences in the prevalence of sarcopenia between CT scan (70.3%) and DEXA (38.7%) were also found in patients with chronic liver disease [24]. In chronic obstructive pulmonary disease, the prevalence of sarcopenia was not different between the two definitions (16.8% vs. 13.1%). [25] Additionally, in patients who underwent curative gastrectomy for gastric cancer, sarcopenia was diagnosed with equal frequency (17.9% vs. 18.9%) with both definitions. [26]. In geriatric patients, the literature is conflicting showing higher [27,28,29] or equal [30,31] rates of sarcopenia with the 2010 compared to the 2019 definition. The predictive utility for complications seems to be different for the 2010 and 2019 definition—in chronic obstructive pulmonary disease both definitions performed well, whereas in the UK biobank cohort the 2010 definition was superior in predicting poorer health outcome. [25,32] The EWGSOP 2019 criteria also identify less cases compared to other internationally used diagnostic criteria in elderly and in cirrhotic patients. [33,34]

## 5. Conclusions

In summary, our study shows for the first time that sarcopenia is less often diagnosed when using the 2019 criteria compared to the 2010 criteria in liver cirrhosis patients as a result of the different starting points and cut-off values within the diagnostic process. The gender imbalance seen in the 2010 definition however seems to be less pronounced with the 2019 definition. Further refinement of cut-off points and assessment tools is necessary to improve diagnostic accuracy. Using specific cut-off points for muscle mass and muscle strength for a particular patient population seems necessary to tailor the definition of sarcopenia to the disease population. Data are lacking, whether the 2010 or 2019 diagnose criteria is better to predict complications, identify poorer prognosis and measure the effect of specific interventions in liver cirrhosis patients.

## Figures and Tables

**Figure 1 nutrients-12-00547-f001:**
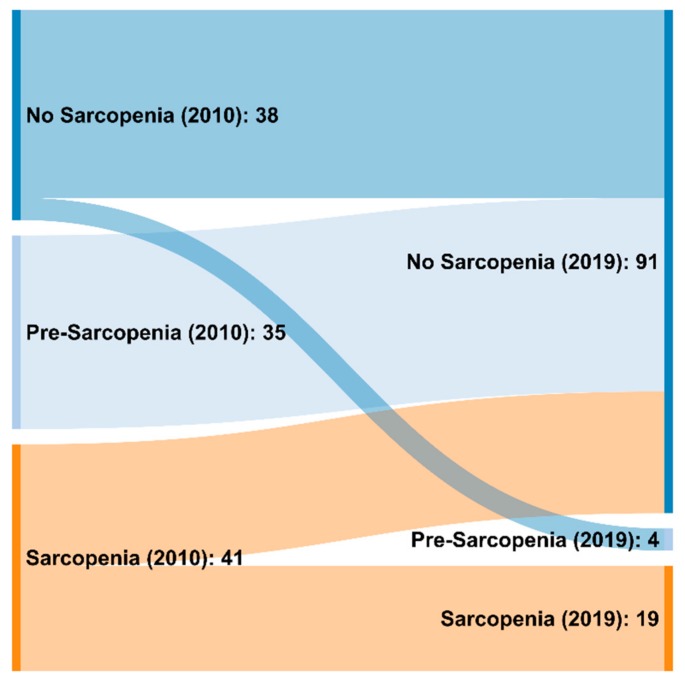
Sankey diagram to visualize and quantify the changes in sarcopenia diagnosis using the EWGSOP 2010 and 2019 criteria.

**Figure 2 nutrients-12-00547-f002:**
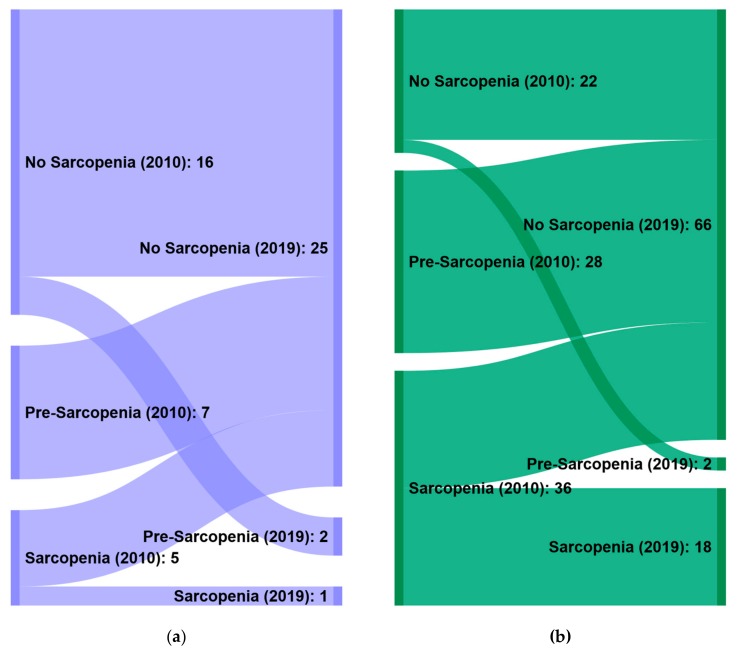
Sankey diagram to visualize and quantify the changes in sarcopenia diagnosis in women (**a**) and men (**b**) using the EWGSOP 2010 and 2019 criteria.

**Table 1 nutrients-12-00547-t001:** Comparison of European Working Group on Sarcopenia in Older People (EWGSOP) 2010 and 2019 diagnosis criteria of sarcopenia.

Stage	2010	2019
Pre-sarcopenia	⇓ Mass	⇓ Strength ²
Sarcopenia	⇓ Mass and ⇓ Strength ^1^ or ⇓ Function	⇓ Strength ² and ⇓ Mass or ⇓ Quality
Severe Sarcopenia	⇓ Mass and ⇓ Strength ^1^ and ⇓ Function	⇓ Strength ² and ⇓ Mass and ⇓ Function

^1^ Cut-off value: <30 kg in men, <20 kg in woman + BMI adapted. ² Cut-off value: <27 kg in men, <16 kg in woman.

**Table 2 nutrients-12-00547-t002:** Baseline characteristics of 114 liver cirrhosis patients.

Characteristics	
Age (year, range)	65 (61.87–65.97)
Sex *m*/*w n* (%)	86/28 (75.4/24.6)
BMI, kg/m^2^	26.83 (26.07–27.95)
Child A/B/C *n* (%)	51/43/20 (44.7/37.7/17.6)
MELD	10.8 (11.54–13.49)
Aetiology: HCV/Alcohol/NASH/Others (%)	17 (14.9)/66 (57.9)/22 (19.3)/9 (7.9)
HCC *n* (%)	54 (47.4)
Diabetes *n* (%)	47 (41.2)
Bilirubin	1.21 (1.89–3.28
Albumin	3.5 (3.37–3.47)
PZ-INR	1.23 (1.24–1.38)
Creatinine	0.9 (0.89–1.09)

Values are median (confidence interval); BMI = Body Mass Index; MELD = Model for End-stage Liver Disease; HCV = Hepatitis C Virus; NASH = Nonalcoholic Steatohepatitis; HCC = Hepatocellular carcinoma; PZ-INR = Prothrombin Time International Normalized Ratio.

**Table 3 nutrients-12-00547-t003:** BMI adaption: Cut-off values hand grip strength 2010.

Gender	BMI	Hand Grip Strength (kg)
male	≤24.0 24.1–26.0 26.1–28.0 >28.0	≤29.0 ≤30.0 ≤30.0 ≤32.0
female	≤23.0 23.1–26.0 26.1–29.0 >29.0	≤17.0 ≤17.3 ≤18.0 ≤21.0

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
