# Peer review of "Sarcopenia and Liver Cirrhosis—Comparison of the European Working Group on Sarcopenia Criteria 2010 and 2019"

_nutrients, 2020, doi:10.3390/nu12020547_

Round 1

Reviewer 1 Report

In a prospective cohort study the authors have compared the presence of sarcopenia in 114 liver cirrhosis patients based on either the 2010 or 2019 European Working Group on Sarcopenia in Older People criteria.

The study is relevant as these criteria is often used is specific disease entities beyond the geriatric field, scientifically sound, well written and interesting. I have a few comments as follows:

Minor points

Line 137, “momentarily” could it rather be “presently” or some other word better describing the essence of the sentence?

Line 138, “widely” should probably be omitted.

Major points

It should be stressed that there is no gold standard definition of sarcopenia in cirrhosis, so what is over- or under diagnosis of sarcopenia?

The relevance of the criteria, being it the 2010 or 2019 criteria, must be their ability to identify patients with a poorer (or for that matter better) prognosis and be a tool to measure the effect of specific intervention.

Was sarcopenia, 2010 or 2019 related to any relevant out-come variables in the 114 patients studied?,

and if not, why so?

Author Response

We thank the reviewer for his/her favourable comments on our manuscript.

  • Line 137, “momentarily” could it rather be “presently” or some other word better describing the essence of the sentence?

“Momentarily” was changed to “presently”

  • Line 138, “widely” should probably be omitted.

“Widely” was deleted

  • It should be stressed that there is no gold standard definition of sarcopenia in cirrhosis, so what is over- or under diagnosis of sarcopenia?

We agree with the reviewer that over- and underdiagnoses is not the correct terminology, since there is no gold standard of diagnosis to compare with and since we cannot correlate our data to outcome data. We therefore now avoid the term over/underdiagnoses and we only talk about the rate of diagnosis. Furthermore, a statement was added, that there is no gold standard for the definition of sarcopenia in patients with liver cirrhosis.

  • Was sarcopenia, 2010 or 2019 related to any relevant out-come variables in the 114 patients studied? and if not, why so?

We agree with the reviewer that a correlation of sarcopenia diagnosis with outcome is of utmost importance. Our study however was planned and conducted as a cross-sectional study, therefore we do not have any outcome data right now. In order to assess outcome data, we would have to seek additional permission by our research ethics committee, which would not be possible to obtain within the narrow time frame of 8 days that was granted to us by the journal to answer to the reviewer’s comments. Furthermore, since the last patient was recruited only 13 months ago, the observation period would be rather short at the moment. But we take up this suggestion and plan to assess outcome of your study population at a later date.

  • The relevance of the criteria, being it the 2010 or 2019 criteria, must be their ability to identify patients with a poorer (or for that matter better) prognosis and be a tool to measure the effect of specific intervention.

We agree with the reviewer and we added a statement in the discussion that data are lacking, whether the 2010 or 2019 diagnose criteria is better to predict complications, identify poorer prognosis and measure the effect of specific interventions in liver cirrhosis patients.

We added the following statement: “Data are lacking, whether the 2010 or 2019 diagnose criteria is better to predict complications, identify poorer prognosis and measure the effect of specific interventions in liver cirrhosis patients.”

Reviewer 2 Report

This is a well written study concerning differences in the rates of diagnosing sarcopenia in patients with liver cirrhosis according to guidelines from the year 2010 and 2019. 

I have one major concern regarding the study. According to the European Working Group on Sarcopenia, the cut-off points for muscle mass should be set as:
"WGSOP recommends use of normative (healthy young adult) rather than other predictive reference populations, with cut-off points at two standard deviations below the mean reference value."
In this study, however, authors used cut-off values from previous studies based on populations of patients with liver cirrhosis, which obviously cannot replace the cut-off points from the general, healthy population, as liver cirrhosis, as a severe chronic disease, leads to catabolic changes and muscle depletion. Since there are no widely accepted cut-off points for muscle mass loss measured on CT scans it may be justified to use alternative cut-off points when examining specific populations. I am however not convinced if it is justified when analyzing different versions of WGSOP guidelines. I think the authors should address this issue in their article. The previous criteria from the year 2010 defined presarcopenia only basing on the muscle mass loss, and the new one from the year 2019 include only muscle strength in the presarcopenic patients, which may influence the results of the study in case of utilization of non optimal cut-off points for muscle mass. 

Author Response

We thank the reviewer for his/her favourable comments on our manuscript.

  • "WGSOP recommends use of normative (healthy young adult) rather than other predictive reference populations, with cut-off points at two standard deviations below the mean reference value."In this study, however, authors used cut-off values from previous studies based on populations of patients with liver cirrhosis, which obviously cannot replace the cut-off points from the general, healthy population, as liver cirrhosis, as a severe chronic disease, leads to catabolic changes and muscle depletion. Since there are no widely accepted cut-off points for muscle mass loss measured on CT scans it may be justified to use alternative cut-off points when examining specific populations. I am however not convinced if it is justified when analyzing different versions of WGSOP guidelines. I think the authors should address this issue in their article.

We agree with the reviewer, that there is no consensus regarding the best cut-off values for muscle mass on CT/MRI scans right now. To clarify the position of the EWGSOP and justify our choice of cut-off levels we added the following statements:

“The cut-offs for low muscle mass are not well defined in both consensus definitions. In our study we therefore decided to use validated liver disease specific cut-offs from a meta-analysis published in 2017.”

“Regarding cut-offs for muscle mass, in the 2010 consensus definition the general statement is made, that a cut-off value of below 2 standard deviations of the mean reference value of healthy young adults should be used, without referencing any specific study or cut-off points for CT/MRI. In the 2019 definition again cut-off values are suggested to be derived from the general healthy population, referencing a study in 420 healthy Caucasians [34], but the necessity of disease specific cut-offs is discussed as well.“

The suggested cut-off values from healthy Caucasians from the reference cited in the EWGSOP 2019 consensus are 38 cm2/m2 in men and 28.9 cm2/m2 in woman. [van der Werf A et al, Eur J Clin Nutr. 2018;72(2):288–296] These cut-off values are significantly lower than the liver cirrhosis specific cut-offs, which are recommended in literature (52.4 cm2/m2 in men and 38.5 cm2/m2 in woman). 76/114 (66.6%) patients would be diagnosed with low muscle mass based on the CT/MRI results using the liver cirrhosis specific cut-offs. When using the cut-offs from healthy controls, only 20/114 (13.8%) patient would be diagnosed with low muscle mass. Using the cut-off values derived from healthy controls therefore leads to a significantly lower rate of diagnosis of low muscle mass (p=0.000, 60.5%). Using the cut-offs from healthy controls would result in only 11/114 (9.6%) patients, that would be diagnosed with sarcopenia compared to 41/114(36%) when using the cirrhosis specific cut-offs with the 2010 definition in our cohort. With the 2019 definition, only 6/114 (5.3%) patients would be diagnosed with sarcopenia with the cut-off values from healthy controls, compared to 19/114 (16.7%) when using the cirrhosis specific cut-offs. In a recent review on sarcopenia [Dasarathy S et al; Journal of hepatology. 2016;65 (6):1232-44], the prevalence of sarcopenia in patients with liver cirrhosis is summarized to be around 30-70% (measured by different methods to quantify muscle mass and function). Therefore, we believe that the usage of the liver cirrhosis cut-off values for our specific patient population is justified. Since we use the same cut-offs for both the 2010 and the 2019 definition, we also believe, that it is valid to make this comparison using the cirrhosis specific cut-off values.

  • The previous criteria from the year 2010 defined presarcopenia only basing on the muscle mass loss, and the new one from the year 2019 include only muscle strength in the presarcopenic patients, which may influence the results of the study in case of utilization of non optimal cut-off points for muscle mass.

We totally agree with the reviewer, that the changes in the starting point of the algorithm from 2010 to 2019 influences the study results. We discuss this in the paper as follows: Results section: “When using the 2019 criteria, 35 patients (30.7%) with reduced muscle mass but preserved muscle strength were not diagnosed as pre-sarcopenia any more compared to the 2010 criteria due to the different starting point of the definition. Four patients (3.5%) who were not diagnosed with sarcopenia according to the 2010 criteria shifted to the pre-sarcopenia group with the 2019 criteria because of a reduction in muscle strength but not in muscle mass.”

Discussion: “Our study shows that this change in the starting point of the definition of sarcopenia (muscle strength versus muscle mass) and in the cut-off values leads to significantly different results in the rate of diagnosis of sarcopenia in liver cirrhosis.“ and „In patients with cirrhosis, muscle strength seems to be preserved longer, while muscle mass is already reduced, leading to significant difference in sarcopenia diagnosis rates when using the 2019 definition, in which muscle strength is the starting point.“

Reviewer 3 Report

There are some difference in sarcopenia criteria 2019 and 2010, for example, muscle strength has replaced the role of muscle mass in pre-sarcopenia diagnosis in 2019. In this article, authors focused on the diagnostic rate of pre-sarcopenia, sarcopenia and severe sarcopenia in liver cirrhosis patients according to the different criteria. They found the rate of pre-sarcopenia was lower based on 2019 criteria than 2010. It is an interesting article, but there are still some research need to be done in future, whether less pre-sarcopenia diagnosis would interfere with the treatment and prognosis in liver cirrhosis patients.

Please replace “cm2/m2” with “cm2/m2” in the whole paper, maybe it is more accurate.

Author Response

We thank the reviewer for his/her favourable comments on our manuscript.

  • Please replace “cm2/m2” with “cm2/m2” in the whole paper, maybe it is more accurate.

cm2/cm2 were replaced with cm2/m2

Round 2

Reviewer 1 Report

Thank you for your clarifications. I have no further comments.

Reviewer 2 Report

The authors revised the manuscript according to the comments and questions and provided additional explanation in the discussion and methods section.